# Mitochondrial Genome Characteristics and Comparative Genomic Analysis of *Spartina alterniflora*

**DOI:** 10.3390/cimb47020107

**Published:** 2025-02-08

**Authors:** Hong Zhu, Chunlei Yue, Hepeng Li

**Affiliations:** 1Zhejiang Academy of Forestry, Hangzhou 310023, China; 84244@163.com (H.Z.); lihpmail@163.com (H.L.); 2Research Centre for Zhejiang Wetland, Hangzhou 310023, China

**Keywords:** *Spartina alterniflora*, mitochondrial genome, comparative analysis, phylogenetic analysis

## Abstract

The mitochondrial genome of *Spartina alterniflora*, an invasive species with significant ecological and economic impacts, was analyzed to provide a theoretical basis for understanding its phylogenetic relationships and molecular biology. Mitochondrial genome sequences of *S. alterniflora* and 23 related species from NCBI were utilized for bioinformatics and comparative genomic analyses. A sliding window analysis identified three genes (*rps2*, *atp9*, and *nad6*) as potential DNA barcodes for species identification. Intracellular gene transfer (IGT) events between mitochondrial and chloroplast genome were detected, highlighting the dynamic nature of genomic evolution. A selective pressure analysis revealed that most protein-coding genes (PCGs) underwent purifying selection (*K_a_*/*K_s_* < 1), while the *nad2* and *ccmB* genes showed signs of positive selection pressure (*K_a_*/*K_s_* > 1), indicating their role in adaptation. A phylogenetic analysis demonstrated a close relationship between *S. alterniflora* and *Eleusine indica*, supported by a collinearity analysis, which suggests environmental convergence. This study provides novel insights into the structural and evolutionary characteristics of the *S. alterniflora* mitochondrial genome, offering valuable genomic resources for future research on invasive species management and evolutionary biology.

## 1. Introduction

*Spartina alterniflora* Loisel. (synonyms: *S. maritima* var. *Alterniflora*, *S. stricta* var. *Alterniflora*, etc.) is a perennial grass of the genus *Spartina* in the Poaceae family. Native to the Atlantic and Gulf coasts of the Americas, this species holds ecological and economical significance. Historically, it has been utilized for coastal stabilization, wetland restoration, and soil erosion control due to its robust root system and rapid growth. Additionally, *S. alterniflora* contributes to carbon sequestration in coastal ecosystems, with its dense biomass acting as a significant carbon sink [1]. However, its invasive proliferation in non-native regions, such as China’s coastal zones, has overshadowed these benefits, posing a significant threat to ecosystems, biodiversity maintenance, and the ecological security of most coastal marsh wetlands [2,3]. Consequently, it has been included in the initial catalog of alien invasive species by the National Environmental Protection Agency of China (https://www.mee.gov.cn/gkml/zj/wj/200910/t20091022_172155.htm/, accessed on 11 April 2024).

Mitochondria are essential organelles in eukaryotic cells, playing a central role in oxidative metabolism, energy synthesis, and physiological processes, such as cellular signal transduction, cell division, differentiation, and apoptosis regulation [4,5]. Plant mitochondrial genomes exhibit remarkable structural diversity and complexity, including polycyclic chromosomes, linear branches, and radial structures [6]. Their sizes ranging from approximately 66 kb to 11.7 Mb and often exist as dynamic, multipartite structures due to frequent recombination mediated by repeat sequences [7,8]. However, the difficulty associated with assembly and high costs of sequencing pose significant challenges to research on plant mitochondria [9,10,11]. With advancement in genomics and sequencing technologies, particularly third-generation sequencing technologies, an increasing number of mitochondrial genomes have been successfully assembled. Species such as *Oryza sativa* L., *Triticum aestivum* L., and *Lolium perenne* L., all belonging to the Poaceae family, have been reported successively. Researchers have systematically investigated research on sequence characteristics, genome structure, and phylogenetic relationships, among other aspects [12,13,14]. For invasive plants like *S. alterniflora*, while the organellar genomes have been sequenced, existing reports have only provided basic information, such as the total mitochondrial genome length (566,328 bp), G + C content, and structural annotation [15,16].

Biological invasions drive environmental changes, potentially endangering local biodiversity, human health, and agricultural and forestry economies [17]. In recent years, mitochondrial genomics has emerged as a valuable research tool, particularly the study of mitochondrial genome dynamics, which provides new insights into the genetic factors underlying invasion success and adaptive evolution [18]. Currently, an in-depth analysis of the mitochondrial genome characteristics and comparative genomics of *S. alterniflora* has not been conducted. Numerous research aspects, including the origin of organellar genomes, the evolution of nuclear–cytoplasmic interactions, structural complexity, and recombination events remain unknown. Therefore, conducting mitochondrial genome research on *S. alterniflora* can enrich the mitochondrial genome information of plants in the Poaceae family and provide essential foundational information for molecular systematics and molecular ecology research on *S. alterniflora*. In this study, we retrieved the mitochondrial genome sequences of *S. alterniflora* and its 23 related species from public databases. Meanwhile, we conducted a comprehensive analysis regarding nucleotide diversity, migration sequences, *K*_a_/*K*_s_ analysis, and comparative genomics. The goal is to explore genomic structural and functional information to deepen our understanding of this species.

## 2. Materials and Methods

### 2.1. Data Source

To elucidate the evolutionary status of the *S. alterniflora* mitochondrial genome, mitochondrial genome sequences of this species and 23 related plants were retrieved from the NCBI Organelle Genome Resources database (http://www.ncbi.nlm.nih.gov/genome/organelle/ accessed on 22 April 2024). The selection criteria were guided by phylogenetic proximity, data availability as well as ecological and functional diversity. A systematic phylogenetic reconstruction was performed using mitochondrial genome data from a single *S. alterniflora* specimen (GenBank accession number: MT471321) publicly available on NCBI, alongside data from 23 related plants. The final dataset included 21 Poales species (20 from the Poaceae family and one from the Cyperaceae family) and an outgroup consisting of three species (one from the Asparagaceae family and two from the Arecaceae family).

### 2.2. Nucleotide Diversity Analysis

The MAFFT software ver. 7.427 (-auto mode) [19] was used to perform a global alignment of shared protein-coding genes (PCGs) from different species. Sliding window analysis was conducted using DnaSP ver. 5 software [20] to calculate the nucleotide diversity (Pi) value for each gene.

### 2.3. Intergenomic Sequence Transfers Analysis

The chloroplast genome sequence (GenBank accession number: MT311317) of *S. alterniflora* were download from the NCBI organelle genome database (https://www.ncbi.nlm.nih.gov/genome/organelle/ accessed on 22 April 2024). The BLAST software was utilized to search for homologous sequences between the chloroplast and mitochondrial genomes, with the filtering criteria set at a match identity of ≥70% and an E-value of ≤1 × 10^−5^. To visually represent the transfer of genomic fragments between the chloroplast and mitochondrial genomes, we used the circos ver. 0.69-5 software (http://circos.ca/software/download/ accessed on 25 April 2024) for visualization.

### 2.4. K_a_/K_s_ Analysis

The ratio of non-synonymous substitutions (*K_a_*) to synonymous substitutions (*K_s_*) was analyzed based on 31 PCGs in the *S. alterniflora* mitochondrial genome, with pairwise comparisons between all species. Gene sequence alignment was conducted using the MAFFT ver. 7.427 software [19], and the *K_a_*/*K_s_* values for each gene were calculated using the *K_a_K_s_*_Calculator ver. 2.0 software [21] with the MLWL model. Finally, a box plot summarizing the *K_a_*/*K_s_* values for each gene was generated.

### 2.5. Phylogenetic Tree Construction

The mitochondrial genomes of 24 representative plants from 18 genera of 4 families in 3 orders were downloaded from NCBI. The coding sequences (CDS) were selected to construct a maximum likelihood phylogenetic tree. The sequences between species were aligned using MAFFT ver. 7.427 software (–auto mode) [19], and the aligned sequences were concatenated. The optimal nucleotide substitution model was determined using jModelTest ver. 2.1.10 software [22]. The GTRGAMMA model was employed in RAxML ver. 8.2.10 software [23] with a Bootstrap value of 1000 to evaluate the maximum likelihood method for building the phylogenetic tree.

### 2.6. Synteny Analysis

The genome sequences were aligned using MUMmer (ver. 4.0.0 beta2) software [24] with the maxmatch parameter to generate dot-plot plots. The x-axis in each box represents the assembled sequence, while the y-axis represents the other sequences. The red lines inside the boxes indicate forward matches, and the blue lines indicate reverse complementary matches.

## 3. Results

### 3.1. Nucleotide Diversity of the S. alterniflora mt Genome

Nucleotide diversity (Pi) can be used to evaluate the variation of nucleic acid sequences among different species. Therefore, selecting regions with higher variation can serve as potential molecular markers. The total number of mutations in 36 genes ranges from 0 to 399, with corresponding Pi values between 0.000 and 0.245, most of which are less than 0.100. Among these, the rRNA gene coding for ribosomal RNA, *rrn5*, has the lowest variability (Pi = 0.000). In contrast, the variable gene coding for the ribosomal small subunit, *rps2*, has the highest variability (Pi = 0.245). The core genes, *atp9* (coding for ATP synthase) with a Pi of 0.070 and *nad6* (coding for NADH dehydrogenase) with a Pi of 0.060, also exhibit slightly higher variability and can be considered as candidate DNA barcoding markers for further research on phylogeny and population genetics of the species (Figure 1). These findings suggest that *rps2*, *atp9*, and *nad6* may serve as candidates for DNA barcoding, enabling rapid and accurate identification of *S. alterniflora* in mixed populations, which is critical for monitoring and controlling its invasive spread.

### 3.2. Intergenomic Sequence Transfers of S. alterniflora

Through a sequence similarity analysis, homologous fragments between the mitochondrial and chloroplast genomes of *S. alterniflora* were detected, revealing extensive interorganellar sequence transfer events. A total of 46 homologous fragments were identified (Figure 2; Table 1), with lengths ranging from 32 to 3203 bp, of which 8 exceed 1000 bp. The total length of the homologous sequences is 28,860 bp, with 28,475 bp located in the chloroplast genome’s repeat regions and 14,962 bp in the mitochondrial genome’s repeat regions. There are 5 protein-coding genes (PCGs) (*ndhJ*, *psaB*, *rpl2*, *rpl23*, and *rps7*) and 12 transfer RNA (tRNA) genes (*trnM*-*CAT*, *trnH*-*GTG*, *trnF*-*GAA*, *trnR-TCT*, *trnS*-*GGA*, *trnP*-*TGG*, *trnC*-*GCA*, *trnW*-*CCA*, *trnN*-*GTT*, *trnN*-*GTT*, and *trnM*-*CAT*) found to be completely located within the homologous sequences between the mitochondrial genome and nuclear genome of *S. alterniflora*. These homologous fragments are referred to as mitochondrial plastid DNAs (MTPTs), signifying chloroplasts to mitochondria transfer [25]. The CP-derived sequences contributed substantially to mitochondrial genomic diversity, consistent with prior studies highlighting species-specific variation in MTPT length and composition. However, the mechanisms driving sequence migration between the genomes and the expression of genes within the migrated sequences remain unknown, warranting further investigation.

### 3.3. Selective Pressure Analysis

Calculating the mean *K*_a_/*K*_s_ can assess the selective pressure in the evolutionary dynamics of protein-coding genes (PCGs) among related species. In the case of neutral selection, *K*_a_/*K*_s_ = 1. When *K*_a_/*K*_s_ > 1, it indicates positive selection, while *K*_a_/*K*_s_ < 1 indicates purifying selection [26]. By comparing the 31 PCGs in the mitochondrial genome of *S. alterniflora* with those of related species, we observed that the *K*_a_/*K*_s_ values ranged from 0.097 to 1.473. Among these, *nad2* (1.473 ± 0.818) and *ccmB* (1.090 ± 0.470) have mean *K*_a_/*K*_s_ values greater than 1, indicating that they are under positive selection, while the remaining 28 PCGs have mean values less than 1, suggesting they are subject to purifying selection (Figure 3).

### 3.4. Phylogenetic Inference

The phylogenetic tree demonstrates a significant divergence between the outgroup and the Poaceae, with a support value of 100%. The 14 taxonomic units within the Poaceae are well clustered. The target species, *S. alterniflora*, forms a distinct subtree with *Eleusine indica* (L.) Gaertn., belonging to the genus *Eleusine*, and is sister to a small clade comprising six genera: *Zea*, *Tripsacum*, *Coix*, *Sorghum*, *Eremochloa*, and *Chrysopogon*. Additionally, *Cyperus esculentus* L. from the genus *Cyperus* is positioned at the base of the Poaceae, characterized by the longest branch length, thereby strongly supporting the separation between the Cyperaceae and Poaceae families (Figure 4).

### 3.5. Sequence Collinearity

In the Poaceae, mitochondrial gene sequence synteny analysis was randomly conducted with *Sorghum bicolor* (L.) Moench, *Triticum aestivum* L., *E. indica*, *Chrysopogon zizanioides* (L.) Roberty, *Agrostis stolonifera* L., and *Eremochloa ophiuroides* (Munro) Hack. alongside *S. alterniflora*. The results of the dot-plot analysis showed that among all pairwise comparisons of the plants, the homologous sequences between *S. alterniflora* and *E. indica* were the longest, accounting for 56.25% and 57.60% of their respective chloroplast genomes. This indicates the highest similarity and a close genetic relationship between the two species (Figure 5).

## 4. Discussion

During the evolution of higher plants, there has been extensive exchange between the mitochondrial and chloroplast genomes, a process known as intracellular gene transfer (IGT) [11,27]. It is hypothesized that IGT occurred as early as the endosymbiotic events that led to the evolution of eukaryotic chloroplasts. This gene transfer process facilitates information exchange and gene recombination between genomes, contributing to the emergence of new genomic structures and functions, thereby enhancing species’ adaptation to varying environmental conditions and survival pressures [28]. IGT is widely regarded as one of the most important driving forces of species evolution [25]. As the number of complete mitochondrial genomes increases, more instances of IGT have been discovered. In this study, approximately 19.15% (25,959 bp) of MTPTs were found in the chloroplast genome, whereas only 3.56% (20,140 bp) of MTPTs were present in the mitochondria, indicating that transfer from chloroplasts is more common than from mitochondria. Sequence transfers between genomes has been ongoing, primarily involving the transfer of non-functional DNA [29]. Although many organelle-derived sequences are inactive or non-functional, there are exceptions, some organelle genes transferred to the host cell nucleus can enhance the host cell’s ability to regulate organelle gene expression [30]. Additionally, tRNA genes are transferred more frequently than PCGs, similar to findings in *Ilex metabaptista* Loes. ex Diels [31] and *Punica granatum* L. [32], indicating that tRNA genes are more conserved in mitochondrial genomes and play an irreplaceable role in mitochondria. In summary, studying IGT is crucial for tracing ancient recombination events and structural variations in plant mitochondrial genomes. However, current explanations of its mechanisms remain at the hypothesis stage. Future research could focus on how the organelle transfer processes in *S. alterniflora* contribute to environmental stress tolerance.

Through a selection pressure analysis, we can gain a deeper understanding of the roles and evolutionary processes of protein-coding genes, providing important clues to unravel the mechanisms underlying species adaptive evolution [31]. The *K*_a_/*K*_s_ statistical results indicate that most variant genes have undergone purifying selection, consistent with studies on species such as *Suaeda glauca* (Bunge) Bunge [33], *Bupleurum chinense* DC. [34], and *Cymbidium ensifolium* (L.) Sw. [35], suggesting that PCGs in the mitochondrial genome are conserved. In contrast, the *nad2* and *ccmB* genes showed signs of positive selection pressure, suggesting their role in adaptation. These findings are consistent with previous studies on other invasive species, indicating that positive selection of energy metabolism-related genes may be a common strategy for successful invasion. The *nad2* gene, also known as mitochondrial NADH dehydrogenase subunit 2, encodes a subunit of NADH dehydrogenase complex I in mitochondria. Its role is to transfer electrons to coenzyme Q within the mitochondrial respiratory chain, thereby generating energy to support cellular metabolism and survival. Studies have shown that mutations or adaptive evolution in *nad2* can enhance the efficiency of oxidative phosphorylation, particularly under environmental stressors such as hypoxia or salinity [36]. The *ccmB* gene encodes a component of the mitochondrial respiratory chain complex, which regulates electron transfer in the electron transport chain and ultimately promotes ATP production, playing a crucial role in maintaining plant respiration and energy generation [37]. Therefore, it is inferred that these two energy metabolism-related genes underwent positive selection during the evolution of *S. alterniflora* to adapt to complex coastal environmental changes, thus sustaining the species’ strong invasive capability and reproductive success. However, further experiments are needed to validate the functions of these key genes through gene annotation and enrichment analyses.

With the ongoing development of high-throughput sequencing technology, the study of phylogenetic relationships has progressively shifted to the genomic level. The topological results of the phylogenetic clustering based on the mitochondrial genome in this study are largely consistent with the classification established by the Angiosperm Phylogeny Group (APG), indicating that utilization information obtained from mitochondrial genomes for plant phylogenetic research is feasible. Wang et al. [16] constructed a maximum likelihood (ML) tree based on 15 mitochondrial genes and suggested that the closest relationship exists between *S. alterniflora* and *S. bicolor*. Building on this foundation, our study expanded the dataset to include 24 sequenced mitochondrial genomes and similarly employed the ML method for phylogenetic reconstruction. The phylogenetic analysis demonstrated a close evolutionary relationship between *S. alterniflora* and *E. indica*, which was further supported by the results of synteny visualized through a dot-plot analysis. This finding is particularly interesting, as it suggests a shared ancestry or convergent evolution under similar environmental pressures (e.g., coastal/ruderal habitats). Both species thrive in disturbed or marginal habitats (e.g., coastal zones for *S. alterniflora* and ruderal environments for *E. indica*). However, due to the lack of sufficiently representative mitochondrial genomes for the genus *Spartina*, it is necessary to obtain additional mitochondrial genomes in the future to better address phylogenetic and evolutionary biology issues concerning this species.

## 5. Conclusions

This study conducted bioinformatics and comparative genomic analyses of *S. alterniflora* and related species based on publicly available mitochondrial genome sequences. The analyses included nucleotide polymorphism, intracellular gene transfer, *K*_a_/*K*_s_ ratios, and a synteny analysis. Additionally, a phylogenetic reconstruction of the coding region sequences from 24 plant mitochondrial genomes further validated the phylogenetic position of *S. alterniflora*. The comprehensive analysis of the structure and function of the mitochondrial genome of *S. alterniflora*, along with the identification of potential molecular markers, contributes to a deeper understanding of the genomic evolution of this invasive species. These findings not only provide valuable genomic resources for future research on invasive species management but also offer potential targets for the development of novel control strategies. Future studies should focus on the functional validation of the identified candidate genes and their role in the invasive success of *S. alterniflora*.

## Figures and Tables

**Figure 1 cimb-47-00107-f001:**
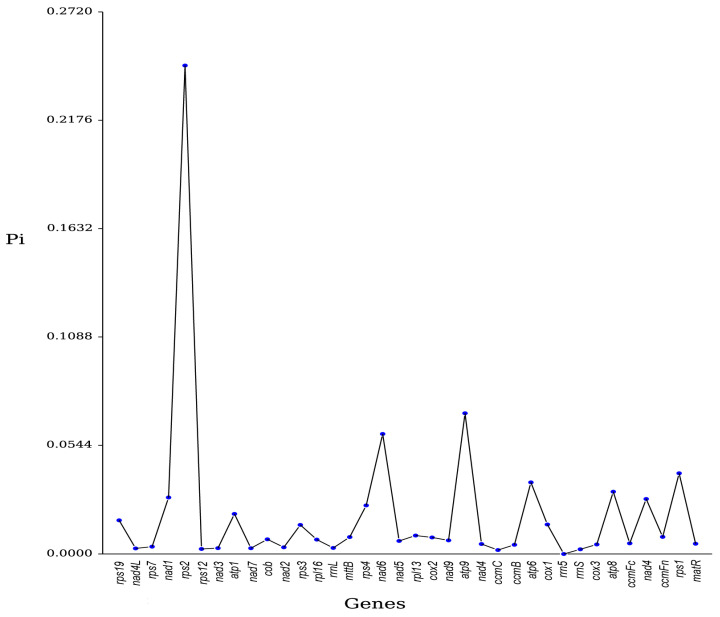
Sliding window analysis for the nucleotide diversity (Pi) of *S. alterniflora*.

**Figure 2 cimb-47-00107-f002:**
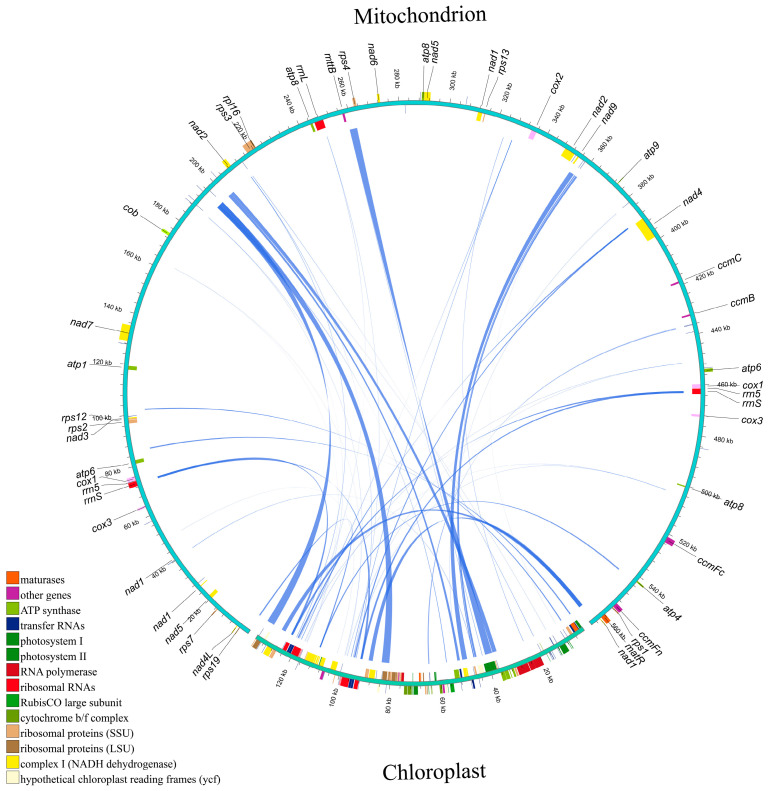
Distribution of homologous fragments between mitochondria and chloroplasts in *S. alterniflora*.

**Figure 3 cimb-47-00107-f003:**
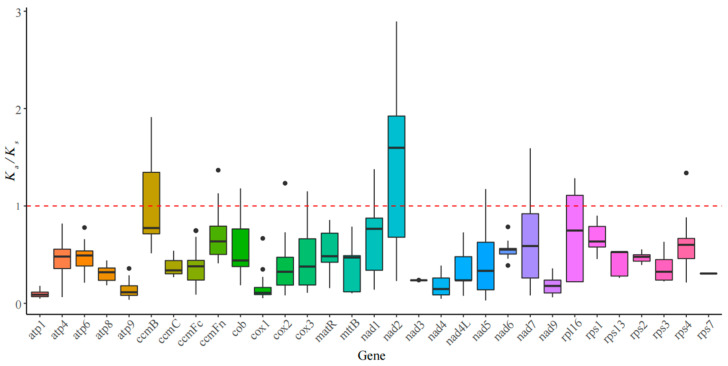
Boxplots of the pairwise *K_a_*/*K_s_* values among every shared mitochondrial gene of *S. alterniflora*.

**Figure 4 cimb-47-00107-f004:**
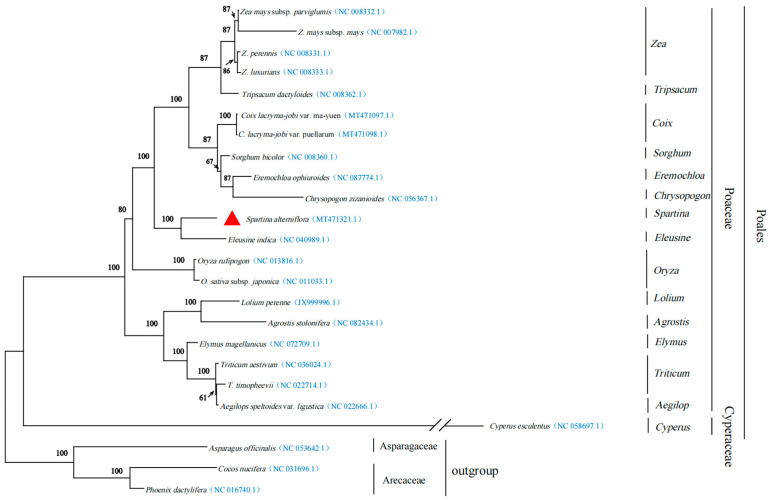
The maximum likelihood phylogenetic tree was constructed based on CDS sequences for 24 species. Numbers at each node represent bootstrap support values. The accession number in blue color following the species name corresponds to the GenBank accession number.

**Figure 5 cimb-47-00107-f005:**
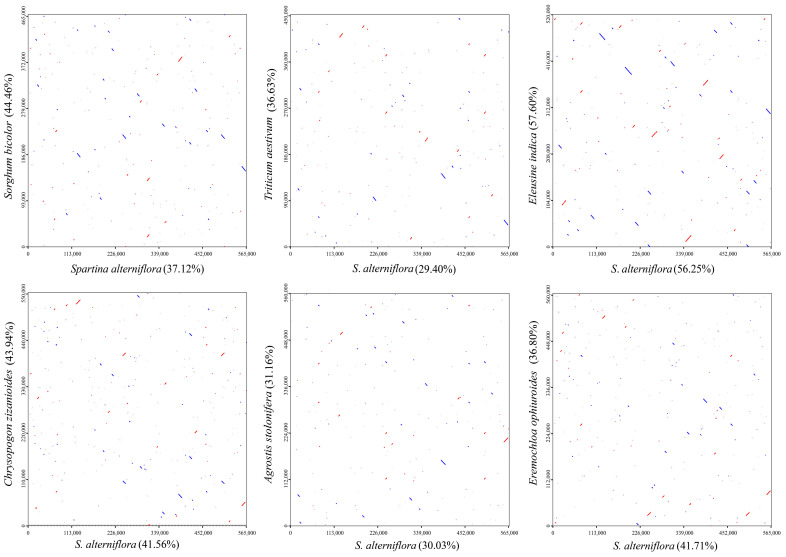
A collinearity analysis of *S. alterniflora* compared to six typical species in Poaceae. The horizontal coordinate in each box indicates the assembled sequence, the vertical coordinate indicates the other sequences, the value in parentheses is the proportion of homologous sequences to the total genome, the red line in the box indicates the forward alignment, and the blue line indicates the reverse complementary alignment.

**Table 1 cimb-47-00107-t001:** Homologous fragments between mitochondria and chloroplasts in *S. alterniflora*.

No.	Aligened Length (bp)	Sequence Identity (%)	Mismatches	Gap Opens	Chloroplast Genome (CP)		Mitochondrial Genome (MT)		Contained Genes
					Start	End	Start	End	
1	3203	99.657	6	1	80,973	84,170	196,986	193,784	*trnM-CAT*; *trnH-GTG*
2	3203	99.657	6	1	132,219	135,416	193,784	196,986	*trnM-CAT*; *trnH-GTG*
3	3007	99.734	2	2	37,454	40,460	258,992	261,992	
4	1709	99.649	2	1	126,698	128,406	565,996	564,292	
5	1709	99.649	2	1	87,983	89,691	564,292	565,996	
6	2166	86.888	147	58	48,194	50,273	357,558	355,444	*trnF-GAA*
7	1297	98.227	20	2	35,467	36,761	203,241	201,946	*trnR-TCT*
8	1034	87.041	84	23	51,282	52,304	201,390	200,396	
9	585	99.658	1	1	95,399	95,982	389,831	389,247	
10	585	99.658	1	1	120,407	120,990	389,247	389,831	
11	825	84.848	70	23	44,149	44,951	359,302	358,511	
12	522	93.678	19	4	622	1137	85,462	85,975	
13	486	93.621	27	2	36,772	37,253	201,889	201,404	
14	532	87.030	51	7	110,463	110,982	592	67	
15	532	87.030	51	7	110,463	110,982	543,296	542,771	
16	453	87.417	41	10	5324	5768	101,801	102,245	
17	258	99.225	1	1	56,844	57,100	194,757	195,014	
18	410	81.463	36	17	45,103	45,487	358,312	357,918	*trnS-GGA*
19	238	89.916	11	8	64,331	64,567	436,134	436,359	*trnP-TGG*
20	889	73.566	180	41	123,201	124,064	73,559	72,701	*rrnS* (partical:43.63%)
21	889	73.566	180	41	92,325	93,188	72,701	73,559	*rrnS* (partical:43.63%)
22	889	73.566	180	41	92,325	93,188	463,240	462,382	*rrnS* (partical:43.63%)
23	889	73.566	180	41	123,201	124,064	462,382	463,240	*rrnS* (partical:43.63%)
24	155	97.419	3	1	41,618	41,772	258,998	258,845	
25	148	97.973	3	0	23,448	23,595	324,214	324,067	
26	226	86.726	29	1	97,053	97,277	327,985	327,760	
27	226	86.726	29	1	119,112	119,336	327,760	327,985	
28	145	93.793	8	1	45,141	45,284	40,212	40,356	*trnS-GGA* (partical:86.52%)
29	182	87.363	11	6	121,318	121,499	212,946	212,777	
30	182	87.363	11	6	94,890	95,071	212,777	212,946	
31	152	87.500	10	6	18,689	18,839	213,562	213,705	*trnC-GCA*
32	93	97.849	1	1	117,548	117,640	450,821	450,730	
33	93	97.849	1	1	98,749	98,841	450,730	450,821	
34	123	87.805	13	2	64,133	64,254	435,907	436,028	*trnW-CCA*
35	81	97.531	2	0	99,799	99,879	188,235	188,155	*trnN-GTT*
36	81	97.531	2	0	116,510	116,590	188,155	188,235	*trnN-GTT*
37	76	92.105	5	1	52,299	52,373	381,683	381,608	*trnM-CAT*
38	97	82.474	17	0	96,206	96,302	248,985	248,889	*rrnL* (partical:2.73%)
39	97	82.474	17	0	120,087	120,183	248,889	248,985	*rrnL* (partical:2.73%)
40	97	82.474	17	0	96,206	96,302	504,491	504,395	
41	97	82.474	17	0	120,087	120,183	504,395	504,491	
42	63	88.889	5	2	12,949	13,011	201,726	201,666	
43	32	100	0	0	99,400	99,431	162,713	162,682	
44	32	100	0	0	116,958	116,989	162,682	162,713	
45	35	97.143	0	1	103,130	103,163	48,940	48,974	
46	37	94.595	1	1	8068	8104	358,125	358,160	*trnS-GGA* (partical:41.38%)
Total	28,860								

## Data Availability

The sequences of complete mitogenomes of *S. alterniflora* can be available in NCBI (accession number: MT311317).

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
