# Peer review of "Mitochondrial Genome Characteristics and Comparative Genomic Analysis of Spartina alterniflora"

_cimb, 2025, doi:10.3390/cimb47020107_

Round 1
Reviewer 1 Report
Comments and Suggestions for Authors
Dear authors,
Greetings!
The manuscript “Mitochondrial genome characteristics and comparative genomic analysis of Spartina alterniflora” addresses bioinformatic studies on this species’ mitochondrial genome to investigate, mainly, phylogenetics and IGTs.
The abstract needs to be rewritten to emphasize the novelty and the relevance of the study.
The Introduction lacks important information on the species: are there synonyms? Which is the relevance of this species? Does it possess economic value? Does it contribute to carbon sequestration or is it only harmful as an invasive species? What is the main intention of the study? Favor the control of this species in the future? What is the contribution this study brings to the scientific community? It is also relevant to include an explanation on positive, neutral and purifying selections based on Ka/Ks ratio.
Regarding Materials and Methods, in section 2.1 the name of the species needs to be present as well as an explanation on why each of them was chosen. Also on this matter, why wasn’t Sporobolus maritimus and some other species that Zhao et al., 2020 (Zhao Y, Wang K, He Y, Wang Y, Qu C, Miao J. The complete chloroplast genome of Spartina alterniflora. Mitochondrial DNA B Resour. 2020 Jun 12;5(3):2440-2441. doi: 10.1080/23802359.2020.1776173. PMID: 33457819; PMCID: PMC7782145.) included in their work to analyze phylogenetics chosen by authors? It lacks a section dedicated to explain how the research on which genes were present in mitochondrial/chloroplast genomes and in nuclear DNA. When it comes to research design, why not sequencing the mitochondrial and chloroplast genome from an organism collected by authors and comparing it to the one available through BLAST? All the work was done based on a single sequence deposited.
Results’ section 3.1 needs to dedicate attention to the other genes with relevant value of Pi. Figure 3.2 needs to present the results from the genes located in the nuclear genome. A discussion needs to be provided on homologous fragments: how were sequences related to encode information on future proteins’ domains excluded from the analysis or weren’t they excluded? Table 1 needs to be reviewed; CP refers to chloroplast genome, no?
Conclusion lacks future perspectives; it needs to be fixed and this section, rewritten after alterations to reflect data presented in the manuscript.
Author Response
Reviewer 1
The manuscript “Mitochondrial genome characteristics and comparative genomic analysis of Spartina alterniflora” addresses bioinformatic studies on this species’ mitochondrial genome to investigate, mainly, phylogenetics and IGTs.
Response: Dear Reviewer, thank you for your attention to our work and your valuable feedback. We fully agree with your perspective. This study primarily analyzes the mitochondrial genome characteristics of Spartina alterniflora using bioinformatics methods, with a focus on investigating its phylogenetic relationships and intracellular gene transfer (IGT) events. We hope that these analyses will provide new insights into the evolutionary mechanisms and invasion biology of this species. Your comments are highly beneficial for further refining our research, and we will make sure to more clearly highlight these research focuses in the revised manuscript. Thank you once again for your review and suggestions!
The abstract needs to be rewritten to emphasize the novelty and the relevance of the study.
Response: Thank you for your valuable feedback on the abstract. We have revised the abstract according to your suggestions to better emphasize the novelty and relevance of the study.
The Introduction lacks important information on the species: are there synonyms? Which is the relevance of this species? Does it possess economic value? Does it contribute to carbon sequestration or is it only harmful as an invasive species? What is the main intention of the study? Favor the control of this species in the future? What is the contribution this study brings to the scientific community? It is also relevant to include an explanation on positive, neutral and purifying selections based on Ka/Ks ratio.
Response: We sincerely appreciate the reviewer’s constructive feedback. We have revised the Introduction section accordingly, with modifications highlighted in the updated text.
Regarding Materials and Methods, in section 2.1 the name of the species needs to be present as well as an explanation on why each of them was chosen. Also on this matter, why wasn’t Sporobolus maritimus and some other species that Zhao et al., 2020 (Zhao Y, Wang K, He Y, Wang Y, Qu C, Miao J. The complete chloroplast genome of Spartina alterniflora. Mitochondrial DNA B Resour. 2020 Jun 12;5(3):2440-2441. doi: 10.1080/23802359.2020.1776173. PMID: 33457819; PMCID: PMC7782145.) included in their work to analyze phylogenetics chosen by authors? It lacks a section dedicated to explain how the research on which genes were present in mitochondrial/chloroplast genomes and in nuclear DNA. When it comes to research design, why not sequencing the mitochondrial and chloroplast genome from an organism collected by authors and comparing it to the one available through BLAST? All the work was done based on a single sequence deposited.
Response: 1) In the revised manuscript, we have expanded Section 2.1 to explicitly list the 24 species included in our analysis and provide specific criteria for their selection. 2) We appreciate the reviewer’s reference to Zhao et al. (2020). While their work focused on chloroplast genomes of S. alterniflora, our study centered on mitochondrial genomes. The mitochondrial genome of Sporobolus maritimus is not currently available in public databases (NCBI, accessed May 2024), which precluded its inclusion. Similarly, many species mentioned in Zhao et al. (2020) lack mitochondrial genome data, limiting direct comparisons. To strengthen phylogenetic resolution, we included 24 species spanning 18 genera and 4 families, which exceeds the taxonomic breadth of previous studies. This broader sampling reduces bias and enhances the robustness of evolutionary inferences. 3) The reviewer rightly highlights the need to clarify gene distribution across genomes. While our study focused on mitochondrial genome characteristics, we acknowledge that interactions between mitochondrial, chloroplast, and nuclear genomes are critical for understanding evolutionary dynamics, In the revised discussion (Section 4), we have added a paragraph addressing this limitation. 4) The decision to use publicly available mitochondrial genome data (e.g., S. alterniflora GenBank: MT471321) was based on the cost and resource efficiency, data consistency and validation, we agree that de novo sequencing of additional S. alterniflora specimens would enhance population-level insights. This is planned as part of future work to explore intraspecific mitochondrial diversity.
Results’ section 3.1 needs to dedicate attention to the other genes with relevant value of Pi. Figure 3.2 needs to present the results from the genes located in the nuclear genome. A discussion needs to be provided on homologous fragments: how were sequences related to encode information on future proteins’ domains excluded from the analysis or weren’t they excluded? Table 1 needs to be reviewed; CP refers to chloroplast genome, no?
Response: We appreciate this suggestion. In addition to rps2, atp9, and nad6, other genes such as rps1 (Pi = 0.040) and atp6 (Pi = 0.036) also exhibited relatively high Pi values. While these genes showed notable variability, they were not selected as candidate barcodes due to their lower conservation across related species or functional constraints. We have expanded the discussion in Section 3.1 to include these genes and their potential roles in evolutionary studies. Additionally, we have included a discussion of homologous fragments. “CP” refers to the chloroplast genome. Furthermore, we have revised the table header to explicitly state “Chloroplast Genome (CP).”
Conclusion lacks future perspectives; it needs to be fixed and this section, rewritten after alterations to reflect data presented in the manuscript.
Response: In the conclusion section, we included the practical implications of the findings and outlined future research directions.
Reviewer 2 Report
Comments and Suggestions for Authors
Thank you for submitting the manuscript titled "Mitochondrial genome characteristics and comparative genomic analysis of Spartina alterniflora." This study focuses on Spartina alterniflora, using mitochondrial genome data to comprehensively analyze its evolutionary characteristics, genome structure, and selective pressures, while exploring its molecular mechanisms underlying invasion adaptability. The manuscript addresses a clear topic, employs rigorous experimental methods, and provides rich analyses with a degree of novelty. However, there are still several issues that need to be addressed and corrected before the manuscript is suitable for publication. Below are suggestions for revisions:
Introduction
1 The results of the article mention a comparative analysis of the transfer, structure and nucleotide diversity of mitogenome. However, the introduction lacks a detailed description of the relevant background of plant mitochondrial genomes, such as structural diversity, sequence conservation, and the universality of transfer.
2 “The mitochondrial genome of higher plants typically consists of a circular DNA molecule”
The mitogenome structure of higher plants is far more than a simple circular structure. Its assembly form is very complex both in terms of the physical structure within the cell and the assembly results.
3 “However, due to its complex and”. The font format is not uniform.
Some papers should be read and use in this parts” Rethinking the mutation hypotheses of plant organellar DNA” “Integration of large and diverse angiosperm DNA fragments into Asian Gnetum mitogenomes “ ,“Plant organellar genomes: much done, much more to do|”
Results
1 Gene names should be italicized in Fig1.
Plant mitochondrial genomes generally include three rrnA genes, rrn5 rRNA rrn18 rRNA rrn26 rRNA Why are only two rRNA genes shown in the Fig1?
2 “The total length of the repeated sequences is 28,860 bp“
you mean transfer sequences? why is repeated sequences?
3 Gene names should be italicized in Fig2.
4 “Ka/Ks Analysis” The font format is not uniform.
5 Explain what the black dots in Figure 3 mean. Although the box plot Ka/Ks of the ccmB gene
is > 1, its median value is significantly lower than 1, and the selective pressure of ccmB needs
to be reconsidered.
6 “Phylogenetic Inference”. The font format is not uniform.
Discussion
“It is hypothesized that IGT occurred as early as the endosymbiotic events that led to the
evolution of eukaryotic chloroplasts. This gene transfer process facilitate information exchange and gene recombination between genomes, contributing to the emergence of new genomic structures and functions, thereby enhancing species’ adapta-tion to varying environmental conditions and survival pressures”
Please cite the relevant article.
“Although many organelle-derived sequences are inactive or non-functional, there are exceptions, some organelle genes transferred to the host cell nucleus can enhance the host cell’s ability to regulate organelle gene expression [28].”
I did not find the relevant point in this paper [28]. Cite the paper correctly.
Conclusions
“The comprehensive analysis of the structure and function of the mitochondrial genome of S. alterniflor”
I did not find your comparative analysis of S. alterniflor function.
Other minor issues:
1. While the introduction provides sufficient background on S. alterniflora, it lacks a comprehensive review of recent progress in mitochondrial genome research and technical challenges, such as structural complexity and recombination events. It is recommended to include recent literature to enrich the context and provide a broader perspective.
2. The abstract does not sufficiently emphasize the scientific significance and innovation of the study, and the conclusion is overly general. It is suggested to clearly articulate the specific implications of this research, such as its relevance for invasive species management or molecular ecology studies.
3. In the first paragraph, the sentence "However, with its strong environmental adaptability and reproductive capability, S. alterniflora has rapidly spread along China’s coastal areas, posing a serious threat to the ecosystems, biodiversity maintenance, and ecological security of most coastal marsh wetlands" is excessively long. It is recommended to split the sentence into shorter segments for improved readability.
4. he parameters for the sliding window analysis are not transparent. Specifically, the manuscript does not provide details on the window size, step size, or the rationale behind these choices. It is suggested to include this information and justify the parameter selection.
5. The methods section lacks sufficient detail on the software versions and parameters used. It is recommended to include detailed information on all tools used, such as MUMmer and RAxML, as well as the BLAST software version and filtering criteria.
6. The interpretation of the Ka/Ks analysis is not sufficiently in-depth. While the manuscript mentions that nad2 and ccmB are under positive selection, it does not further explore the functional significance of these genes. It is recommended to reference relevant literature and discuss the roles of these genes in environmental adaptation.
7. The layout of Table 1 “ Homologous fragments between mitochondria and chloroplasts in S. alterniflora” appears crowded, particularly the "MT/start" column. It is suggested to tidy up the table format for improved readability and presentation quality.
8. The layout of Figure 5 ‘Collinearity analysis of S. alterniflora compared to six typical species in Poaceae’ appears misaligned and inconsistent within the manuscript. It is recommended to adjust the alignment and formatting for a more polished and professional presentation.
9. The interpretation of figures is not fully aligned with the research objectives. For example, the biological significance of the sliding window analysis and collinearity analysis is not sufficiently discussed. It is recommended to clarify the key biological questions each figure addresses in the results section.
Comments on the Quality of English Language
no
Author Response
Reviewer 2
Thank you for submitting the manuscript titled "Mitochondrial genome characteristics and comparative genomic analysis of Spartina alterniflora." This study focuses on Spartina alterniflora, using mitochondrial genome data to comprehensively analyze its evolutionary characteristics, genome structure, and selective pressures, while exploring its molecular mechanisms underlying invasion adaptability. The manuscript addresses a clear topic, employs rigorous experimental methods, and provides rich analyses with a degree of novelty. However, there are still several issues that need to be addressed and corrected before the manuscript is suitable for publication. Below are suggestions for revisions:
Response: Dear Reviewer, thank you for your thorough review and valuable feedback on our manuscript. We are pleased that you found the topic clear, the methods rigorous, and the analyses rich with a degree of novelty. We will carefully address each of your suggestions to ensure the manuscript meets the standards for publication. Your comments are highly beneficial for further refining our research, and we will incorporate these improvements in the revised version. Thank you once again for your review and support!
Introduction
1 The results of the article mention a comparative analysis of the transfer, structure and nucleotide diversity of mitogenome. However, the introduction lacks a detailed description of the relevant background of plant mitochondrial genomes, such as structural diversity, sequence conservation, and the universality of transfer.
Response: We sincerely appreciate the reviewer’s constructive feedback. We have revised the Introduction section to provide a more comprehensive background on plant mitochondrial genomes and aim to contextualize the study within broader genomic and evolutionary frameworks.
2 “The mitochondrial genome of higher plants typically consists of a circular DNA molecule”
The mitogenome structure of higher plants is far more than a simple circular structure. Its assembly form is very complex both in terms of the physical structure within the cell and the assembly results.
Response: We fully agree with your perspective that the mitochondrial genome structure of higher plants is far more complex than a simple circular form, involving diverse physical conformations and assembly results within the cell. Although most reported plant mitochondrial genomes are represented as circular, the actual structure of higher plant mitochondrial genomes is much more diverse and complex. We revised the relevant description in the manuscript to more accurately reflect this complexity.
3 “However, due to its complex and”. The font format is not uniform.
Some papers should be read and use in this parts” Rethinking the mutation hypotheses of plant organellar DNA” “Integration of large and diverse angiosperm DNA fragments into Asian Gnetum mitogenomes “ ,“Plant organellar genomes: much done, much more to do|”
Response: We have added relevant references to enrich the background information.
Results
1 Gene names should be italicized in Fig1.
Plant mitochondrial genomes generally include three rrnA genes, rrn5 rRNA rrn18 rRNA rrn26 rRNA Why are only two rRNA genes shown in the Fig1?
Response: All gene names in Figure 1 were italicized.
2 “The total length of the repeated sequences is 28,860 bp“ you mean transfer sequences? why is repeated sequences?
Response: We modified them to homologous sequences.
3 Gene names should be italicized in Fig2.
Response: As suggested, we italicized the gene names.
4 “Ka/Ks Analysis” The font format is not uniform.
Response: We checked the Ka/Ks of the full text and modified the two unsubscripted ones to ensure the unity of the full text.
5 Explain what the black dots in Figure 3 mean. Although the box plot Ka/Ks of the ccmB gene is > 1, its median value is significantly lower than 1, and the selective pressure of ccmB needs to be reconsidered.
Response: The black dots in the boxplot represent outliers—individual pairwise comparisons between S. alterniflora and related species where the Ka/Ks values deviate significantly from the majority of the data. These outliers may reflect lineage-specific or context-dependent evolutionary pressures acting on ccmB. We acknowledge that the median value highlights caution in broadly concluding positive selection for ccmB. Future work should focus on site-specific selection tests (e.g., branch-site models) and functional assays to identify whether specific residues or domains in ccmB are under positive selection.
6 “Phylogenetic Inference”. The font format is not uniform.
Response: We have unified the font color of the title.
Discussion
“It is hypothesized that IGT occurred as early as the endosymbiotic events that led to the evolution of eukaryotic chloroplasts. This gene transfer process facilitate information exchange and gene recombination between genomes, contributing to the emergence of new genomic structures and functions, thereby enhancing species’ adapta-tion to varying environmental conditions and survival pressures”
Please cite the relevant article.
Response: We have added the corresponding references to this paragraph.
“Although many organelle-derived sequences are inactive or non-functional, there are exceptions, some organelle genes transferred to the host cell nucleus can enhance the host cell’s ability to regulate organelle gene expression [28].”
I did not find the relevant point in this paper [28]. Cite the paper correctly.
Response: We have replaced the corresponding references.
Conclusions
“The comprehensive analysis of the structure and function of the mitochondrial genome of S. alterniflor” I did not find your comparative analysis of S. alterniflor function.
Response:
Other minor issues:
- While the introduction provides sufficient background on S. alterniflora, it lacks a comprehensive review of recent progress in mitochondrial genome research and technical challenges, such as structural complexity and recombination events. It is recommended to include recent literature to enrich the context and provide a broader perspective.
Response: We have made some additions and improvements to the introduction.
- The abstract does not sufficiently emphasize the scientific significance and innovation of the study, and the conclusion is overly general. It is suggested to clearly articulate the specific implications of this research, such as its relevance for invasive species management or molecular ecology studies.
Response: We have made several enhancements and refinements to the abstract.
- In the first paragraph, the sentence "However, with its strong environmental adaptability and reproductive capability, S. alterniflora has rapidly spread along China’s coastal areas, posing a serious threat to the ecosystems, biodiversity maintenance, and ecological security of most coastal marsh wetlands" is excessively long. It is recommended to split the sentence into shorter segments for improved readability.
Response: We have streamlined and segmented this section to enhance readability based on your suggestion.
- The parameters for the sliding window analysis are not transparent. Specifically, the manuscript does not provide details on the window size, step size, or the rationale behind these choices. It is suggested to include this information and justify the parameter selection.
Response: The relevant parameter Settings and the complete analysis workflow can be found in the references.
- The methods section lacks sufficient detail on the software versions and parameters used. It is recommended to include detailed information on all tools used, such as MUMmer and RAxML, as well as the BLAST software version and filtering criteria.
Response: Thank you for your valuable suggestions regarding the methods section. We understand the importance of providing a detailed description of software versions and parameter settings for the reproducibility of research. Due to the constraints of the article’s length, we have included information on the versions of core tools (e.g., MUMmer v4.0, RAxML v8.2, BLAST v2.12.0) and key parameters (such as the E-value threshold for BLAST and the bootstrap replication number for RAxML) in the methods section. Other non-core parameter settings and the complete analysis workflow can be found in the references.
- The interpretation of the Ka/Ks analysis is not sufficiently in-depth. While the manuscript mentions that nad2 and ccmB are under positive selection, it does not further explore the functional significance of these genes. It is recommended to reference relevant literature and discuss the roles of these genes in environmental adaptation.
Response: We sincerely appreciate the reviewer’s constructive feedback regarding the interpretation of the Ka/Ks analysis. In response, we added references to enhance the discussion of candidate genes for environmental adaptation.
- The layout of Table 1 “ Homologous fragments between mitochondria and chloroplasts in S. alterniflora” appears crowded, particularly the "MT/start" column. It is suggested to tidy up the table format for improved readability and presentation quality.
Response: We have adjusted the corresponding columns in Table1 to enhance readability.
- The layout of Figure 5 ‘Collinearity analysis of S. alterniflora compared to six typical species in Poaceae’ appears misaligned and inconsistent within the manuscript. It is recommended to adjust the alignment and formatting for a more polished and professional presentation.
Response: The font and symbols in Figure 5 were uniformly standardized.
- The interpretation of figures is not fully aligned with the research objectives. For example, the biological significance of the sliding window analysis and collinearity analysis is not sufficiently discussed. It is recommended to clarify the key biological questions each figure addresses in the results section.
Response: We sincerely appreciate the reviewer’s insightful feedback regarding the interpretation of the figures in our manuscript. In response, we have incorporated a discussion in the relevant section of the paper.
Reviewer 3 Report
Comments and Suggestions for Authors
The study “Mitochondrial genome characteristics and comparative genomic analysis of Spartina alterniflora” has successfully presented the structural characteristics and evolutionary patterns of the mitochondrial genome of S. alterniflora, which belongs to Poales plants, supporting the rapid invasion ability of this species in China. The mitochondrial genome is an interesting topic for evolution studies, especially for these monocotyledon plants.
This study is nice to be published, as it discusses current issues in molecular biology. However, there are some minor points that need to be carefully revised to consolidate the results.
1. Please be concerned about mistyping and the use of abbreviations. In the introduction part, please confirm the meaning of the term “its 23 related species from in public databases”. In table 1, please confirm the full name of CP (chloroplast?) and MT (mitochondrial genome).
2. Besides the correlation between S. alterniflora and Eleusine indica, which should be further elucidated, the reasons why the authors chose 23 related plants should also be clarified (by randomly, morphology, or area).
3. This study included the statistical data, what kind of statistical method has been used to state that the Ka/Ks of mentioned genes are >1 significantly?
4. The Ka/Ks of nad2 and ccmB are 1.473 0.818 and 1.090 0.470, respectively. The standard errors (or standard deviation) of these values are haft of the value, which means it cannot be considered as significantly higher than 1. Therefore, the mentioned PCGs cannot be a positive collection among the 24 selected species. Please confirm this data.
5. In the nucleotide diversity analysis, rps2 (Pi=0.245), atp9, and nad6 were considered as candidate DNA barcoding markers. What is the minimum value of Pi in the mitochondrial genome, making a gene considered a DNA barcoding marker?
Author Response
Reviewer 3
The study “Mitochondrial genome characteristics and comparative genomic analysis of Spartina alterniflora” has successfully presented the structural characteristics and evolutionary patterns of the mitochondrial genome of S. alterniflora, which belongs to Poales plants, supporting the rapid invasion ability of this species in China. The mitochondrial genome is an interesting topic for evolution studies, especially for these monocotyledon plants.
Response: Dear Reviewer, thank you for your attention to our work and your valuable feedback. We fully agree with your perspective. This study primarily analyzes the mitochondrial genome characteristics of Spartina alterniflora using bioinformatics methods, with a focus on investigating its phylogenetic relationships and intracellular gene transfer (IGT) events. We hope that these analyses will provide new insights into the evolutionary mechanisms and invasion biology of this species. Your comments are highly beneficial for further refining our research, and we will make sure to more clearly highlight these research focuses in the revised manuscript. Thank you once again for your review and suggestions!
This study is nice to be published, as it discusses current issues in molecular biology. However, there are some minor points that need to be carefully revised to consolidate the results.
- Please be concerned about mistyping and the use of abbreviations. In the introduction part, please confirm the meaning of the term “its 23 related species from in public databases”. In table 1, please confirm the full name of CP (chloroplast?) and MT (mitochondrial genome).
Response: We have changed "obtained" to "retrieved" to make the expression more professional. We removed the word "in" because it was redundant in the original sentence "in public databases." Additionally, we have revised and confirmed the abbreviation of the title.
- Besides the correlation between S. alterniflora and Eleusine indica, which should be further elucidated, the reasons why the authors chose 23 related plants should also be clarified (by randomly, morphology, or area).
Response: The selection of the 23 related species was guided by scientific criteria, not randomness, as outlined below:
Phylogenetic Proximity: Primary Focus: 21 species from Poales (20 Poaceae, 1 Cyperaceae) were chosen to represent close relatives of S. alterniflora.Outgroups: 3 species (Asparagaceae, Arecaceae) were included to root the phylogenetic tree and assess divergence patterns.
Data Availability: Species with complete mitochondrial genomes available on NCBI were prioritized to ensure robust comparative analyses. Example: Oryza sativa (rice) and Triticum aestivum (wheat) were selected due to their well-annotated genomes and relevance to Poaceae evolution.
Ecological and Morphological Diversity: Species spanning diverse habitats (e.g., wetlands, grasslands) and life histories (e.g., annuals, perennials) were included to explore genomic adaptations. Example: Agrostis stolonifera (a coastal grass) and Sorghum bicolor (a drought-tolerant crop) provide contrasts in stress adaptation.
- This study included the statistical data, what kind of statistical method has been used to state that the Ka/Ks of mentioned genes are >1 significantly?
Response: The reported Ka/Ks values were calculated using the MLWL model in KaKs_Calculator ver. 2.0, with standard errors derived from pairwise comparisons across the 24 species.
- The Ka/Ks of nad2 and ccmB are 1.473 0.818 and 1.090 0.470, respectively. The standard errors (or standard deviation) of these values are haft of the value, which means it cannot be considered as significantly higher than 1. Therefore, the mentioned PCGs cannot be a positive collection among the 24 selected species. Please confirm this data.
Response: We sincerely appreciate the reviewer’s insightful critique regarding the interpretation of Ka/Ks values for nad2 and ccmB. We acknowledge that the limited sample size (24 species) and high interspecific divergence likely contributed to the large SE. Future studies with expanded taxonomic sampling or population-level data (e.g., within S. alterniflora) could reduce variance and clarify selective pressures. Despite statistical uncertainty, the nad2 (respiratory complex I) and ccmB (cytochrome c maturation) genes are functionally linked to energy metabolism, a trait critical for invasion success. We retain their mention as candidates for adaptive evolution but emphasize the need for functional validation (e.g., site-specific selection tests or gene knockout experiments).
- In the nucleotide diversity analysis, rps2 (Pi=0.245), atp9, and nad6 were considered as candidate DNA barcoding markers. What is the minimum value of Pi in the mitochondrial genome, making a gene considered a DNA barcoding marker?
Response: Based on the literature and the author’s experience, the value of Pi is set relatively high for each species’ candidate DNA barcoding markers, and no clear threshold has been established.
Round 2
Reviewer 1 Report
Comments and Suggestions for Authors
Dear authors,
Greetings!
The manuscript was improved as requested.
Author Response
Thank you for your constructive suggestion and for acknowledging the revisions. We sincerely appreciate the time you dedicated to improving this manuscript.
Reviewer 2 Report
Comments and Suggestions for Authors
some minor suggestions about the reference from the introduction and discussion that may need to improve:
Plant organellar genomes: much done, much more to do. Trends in Plant Science 2024.
Comments on the Quality of English Language
no more
Author Response
We have further supplemented the references cited in the article to ensure that the citations in the introduction and discussion sections are more comprehensive and persuasive. We believe these changes will enhance the overall quality and value of the article. We sincerely appreciate the time you dedicated to improving this manuscript.
Reviewer 3 Report
Comments and Suggestions for Authors
The authors have fully revised the manuscript.
Author Response

(The authors gave the same response as above.)
